# lncRNA requirements for mouse acute myeloid leukemia and normal differentiation

M Joaquina Delás[1,2†], Leah R Sabin[2†], Egor Dolzhenko[3†], Simon RV Knott[1,2], Ester Munera Maravilla[1], Benjamin T Jackson[1], Sophia A Wild[1,4], Tatjana Kovacevic[1], Eva Maria Stork[1], Meng Zhou[3], Nicolas Erard[1], Emily Lee[2], David R Kelley[5], Mareike Roth[6], Inês AM Barbosa[6], Johannes Zuber[6], John L Rinn[5‡], Andrew D Smith[3*], Gregory J Hannon[1,2,7*]

[1]Cancer Research UK Cambridge Institute, Li Ka Shing Centre, University of Cambridge, Cambridge, United Kingdom; [2]Watson School of Biological Sciences, Howard Hughes Medical Institute, Cold Spring Harbor Laboratory, New York, United States; [3]Molecular and Computational Biology, University of Southern California, Los Angeles, United States; [4]German Cancer Research Center, Heidelberg, Germany; [5]Department of Stem Cell and Regenerative Biology, Harvard University, Cambridge, United States; [6]Research Institute of Molecular Pathology, Vienna Biocenter, Vienna, Austria; [7]New York Genome Center, New York, United States

**\*For correspondence:**
andrewds@usc.edu (ADS);
greg.hannon@cruk.cam.ac.uk
(GJH)

[†]These authors contributed
equally to this work

**Present address:** [‡]Biofrontiers
Department of Biochemistry,
University of Colorado Boulder,
Boulder, United States

**Competing interests:** The
authors declare that no
competing interests exist.

**Reviewing editor:** Juan
Valcárcel, Centre de Regulació
Genòmica (CRG), Barcelona,
Spain

**Abstract** A substantial fraction of the genome is transcribed in a cell-type-specific manner, producing long non-coding RNAs (lncRNAs), rather than protein-coding transcripts. Here, we systematically characterize transcriptional dynamics during hematopoiesis and in hematological malignancies. Our analysis of annotated and de novo assembled lncRNAs showed many are regulated during differentiation and mis-regulated in disease. We assessed lncRNA function via an in vivo RNAi screen in a model of acute myeloid leukemia. This identified several lncRNAs essential for leukemia maintenance, and found that a number act by promoting leukemia stem cell signatures. Leukemia blasts show a myeloid differentiation phenotype when these lncRNAs were depleted, and our data indicates that this effect is mediated via effects on the MYC oncogene. Bone marrow reconstitutions showed that a lncRNA expressed across all progenitors was required for the myeloid lineage, whereas the other leukemia-induced lncRNAs were dispensable in the normal setting.
DOI: https://doi.org/10.7554/eLife.25607.001

## Introduction

Long noncoding RNAs (lncRNAs) have emerged as an additional layer of regulation of gene expression (*Rinn and Chang, 2012*). Although their definition is rather arbitrary – transcripts longer than 200 bp with little or no evidence of protein coding capacity – their reported functions are essential and diverse (*Wang and Chang, 2011*). A number of different roles have been ascribed to lncRNAs during differentiation (*Fatica and Bozzoni, 2014*) yet the function of most lncRNAs remains unexplored. Their cell-type-specific expression has encouraged the study of lncRNA function during development, where lncRNAs important for dendritic cell specification, epidermal, and cardiac differentiation have been identified (*Grote et al., 2013*; *Klattenhoff et al., 2013*; *Kretz et al., 2013*; *Wang et al., 2014*). Several recent large-scale cataloging efforts have highlighted how lncRNAs are

also differentially expressed in human cancers (*Du et al., 2013*; *Iyer et al., 2015*; *Yan et al., 2015*), with a few being the subject of more detailed mechanistic studies. In breast cancer models, *HOTAIR* has been shown to promote metastasis through re-location of PRC2 (*Gupta et al., 2010*), and *PVT1* expression correlates with MYC protein levels and influences its stability (*Tseng et al., 2014*). In T cell acute lymphoblastic leukemia (T-ALL), expression analysis revealed many Notch-regulated lncRNAs. Amongst them, *LUNAR* was shown to act as an enhancer-like RNA, activating expression of *IGF1R* (*Trimarchi et al., 2014*).

Development of T-ALL is not the only aspect of hematopoiesis regulated by lncRNAs. *lncRNA-EPS* promotes survival and inhibits apoptosis in murine fetal erythroblasts (*Hu et al., 2011*) and represses key immune genes in macrophages to restrain inflammation in vivo (*Atianand et al., 2016*). In humans, *lncRNA-DC* is required for dendritic cell differentiation through its binding to STAT3 (*Wang et al., 2014*). Global analyses showed GENCODE-annotated lncRNAs to be regulated in mouse early hematopoietic progenitors (*Cabezas-Wallscheid et al., 2014*). Further studies have carried out de novo assemblies of the lncRNA repertoire in murine erythroid (*Alvarez-Dominguez et al., 2014*), erythro-megakaryocytic differentiation (*Paralkar et al., 2014*), and hematopoietic stem cells (HSCs), where two novel lncRNAs were characterized and found to regulate HSC function (*Luo et al., 2015*).

A comprehensive analysis of lncRNA dynamics through normal and malignant hematopoiesis has yet to be reported. The murine hematopoietic system is a very well characterized model of stem and progenitor cell differentiation. Decades of research have provided information on many of the genes that govern the maintenance of HSCs, as well as downstream differentiation events. Many of the same transcription factors required for progenitor self renewal and specification are involved in malignant transformation (*Krivtsov et al., 2006*). This makes hematopoiesis an excellent context for a systematic comparison of lncRNA function in normal development and cancer.

We sought to identify de novo the lncRNAs expressed during the differentiation of both the myeloid and lymphoid hematopoietic lineages, as well as those lncRNAs that are characteristic of transformed cells, using models of acute myeloid leukemia (AML) and B-cell lymphoma. This transcriptome analysis revealed a large number of lncRNAs that are tightly regulated during hematopoietic cell-fate choices. As a first approach to identify functionally relevant lncRNAs, we decided to focus on an in vivo model of murine AML.

AML is often driven by fusion transcription factors or chromatin modifiers, such as MLL-AF9, that maintain an aberrant transcriptional landscape in transformed cells. Consequently, interfering with these chromatin-modifying complexes can lead to a substantial reduction in proliferation of these cancer cells (*Dawson et al., 2011*; *Roe et al., 2015*; *Shi et al., 2013*; *Zuber et al., 2011c*). Interestingly, one of the reported functions for lncRNAs is the regulation of gene activity through interactions on chromatin. For example, lncRNAs *HOTTIP* and *HoxBLinc*, have been shown to activate expression of Hox genes by mediating recruitment of histone methyltransferase complexes WDR5-MLL and Setd1a/MLL1, respectively (*Deng et al., 2016*; *Wang et al., 2011*), and *HOTAIR* regulates the chromatin landscape via recruitment of PRC2 and the LSD1/CoREST/REST complexes (*Rinn et al., 2007*; *Tsai et al., 2010*). As lncRNAs have been associated with chromatin regulation, it seemed possible that these might play a role in enforcing the aberrant transcriptional landscape in AML.

Our systematic analysis of lncRNA transcription in hematopoietic differentiation and AML revealed large numbers of lncRNAs misregulated in diseased or shared between AML and normal cell types. To test whether lncRNAs could regulate the disease state, we used the MLL-AF9-driven AML model to perform an in vivo shRNA screen. We chose a set of 120 lncRNAs with varying expression patterns and levels, and identified several lncRNAs required for maintaining leukemia proliferation in vitro and in vivo. Silencing of several lncRNAs needed for AML proliferation in vitro resulted in patterns of differentiation that mimicked those that occurred upon reduction in the activity of well-established oncogenic drivers. We performed bone marrow reconstitutions for the three lncRNAs showing this phenotype and found that the lncRNA with expression across multiple hematopoietic progenitors to be required for the myeloid lineage, while the two leukemia-induced lncRNAs were dispensable in the normal setting. Collectively, this study serves as a framework for further mechanistic studies of the roles of lncRNAs in hematological malignancies and normal differentiation.

## Results

### A comprehensive catalog of lncRNAs in the hematopoietic system

To characterize the lncRNA repertoire and assess how different non-coding transcripts are regulated during hematopoietic differentiation and disease, we produced a comprehensive catalog of murine hematopoietic lncRNAs. We performed deep RNA sequencing (RNAseq) using 11 cell types representing different stages of hematopoietic differentiation, ranging from long-term hematopoietic stem cells (LT-HSC) to differentiated cell types and blood cancers (*Figure 1A*). Each library was sequenced and mapped to the mm10 genome assembly, with an average of 100 million uniquely mapped reads. We performed de novo transcriptome assembly for each library using cufflinks (*Trapnell et al., 2010*), with the GENCODE annotation (*Harrow et al., 2006*) as a reference transcriptome. Assembled gene models that overlapped with GENCODE coding gene models in the same orientation were discarded. Within each gene model, we required each transcript isoform to be independently assembled from two different libraries, and we filtered based on coding potential (*Figure 1B*, Materials and methods).

We observed a substantial overlap between our lncRNA genes and GENCODE lncRNAs, as well as the lncRNA catalogs from megakaryocyte-erythroid progenitors (MEP) differentiated in vitro (*Paralkar et al., 2014*), erythrocyte differentiation (*Alvarez-Dominguez et al., 2014*), and HSC, B cells, and Gr1 myeloid cells (*Luo et al., 2015*). This validated our assembly pipeline. Interestingly, over half of the lncRNAs assembled were unique to our study, likely due to our sequencing depth and the number of new cell types included (*Figure 1C*). We next used ATACseq data to assess chromatin accessibility at these lncRNA loci. These datasets included some of the same cell types that we analyzed, including the oligopotent myeloid progenitors, hematopoietic stem and progenitor cells (LSK) fraction (less pure than our LT-HSC), and differentiated cell types from both myeloid and lymphoid lineages (*Lara-Astiaso et al., 2014*). A meta analysis of transcriptional start sites (TSSs) within our full lncRNA catalog revealed a correlated open chromatin signal in every cell type with ATACseq data. The number of lncRNAs in our catalog that showed enrichment varied between cell types (*Figure 1D*), which is to be expected given that each expresses only a subset of lncRNAs. We performed the same analysis for the start of the second exon as a control region, and no signal above background was observed.

LncRNAs can have a number of different relationships to their neighboring protein coding genes. They can fall in intergenic regions, be divergently or convergently transcribed, they can overlap in antisense orientation (interspersed), or they can have the same orientation as the neighboring gene, upstream or downstream. To address the possibility that the open chromatin signatures we observed were exclusively the result of regulatory regions being shared between lncRNAs and neighboring protein-coding genes, we performed our analysis independently for each category of lncRNA defined above. Irrespective of their relationship to surrounding protein-coding genes, our assembled lncRNAs showed enrichment in ATACseq signal at their presumed TSS in at least one cell type (*Figure 1—figure supplement 1* ).

Using DESeq2 (*Love et al., 2014*), we performed principal component analysis (PCA) based on the 500 most variable protein-coding genes or lncRNAs from our catalog. The lymphoid differentiated cell types CD3, PreB, and ProB clustered together and clearly separated from the myeloid differentiated cell type Gr1 and the progenitors. Despite having very different functional properties, oligopotent progenitors and long-term repopulating hematopoietic stem cells (LT-HSCs) are found in close proximity, indicating that they share some transcriptional programs. Interestingly, the closest progenitors to the lymphoid differentiated cluster were the common lymphoid progenitors (CLPs). The acute myeloid leukemia samples, both in vivo (AML) and in vitro (RN2 cell line), clustered closest to the granulocyte macrophage progenitor (GMP) population, consistent with previous reports for this AML model (*Krivtsov et al., 2006*) (*Figure 1E*, left). PCA based on lncRNA rather than coding gene expression replicated all the aforementioned features, indicating that lncRNA expression patterns are overall very similar to those of coding genes (*Figure 1E*, right). To confirm the identity of our sorted progenitor populations, we additionally compared the expression signatures from our data with previously published microarray data for these same cell types (*Gazit et al., 2013*) (*Figure 1—figure supplement 1B*). In general, our data was in substantial agreement with prior microarray datasets with the exception of those from CLPs. Notably, CLPs were the one cell type where our staining strategy and isolation protocols (*Figure 1—figure supplement 2B*) differed from those

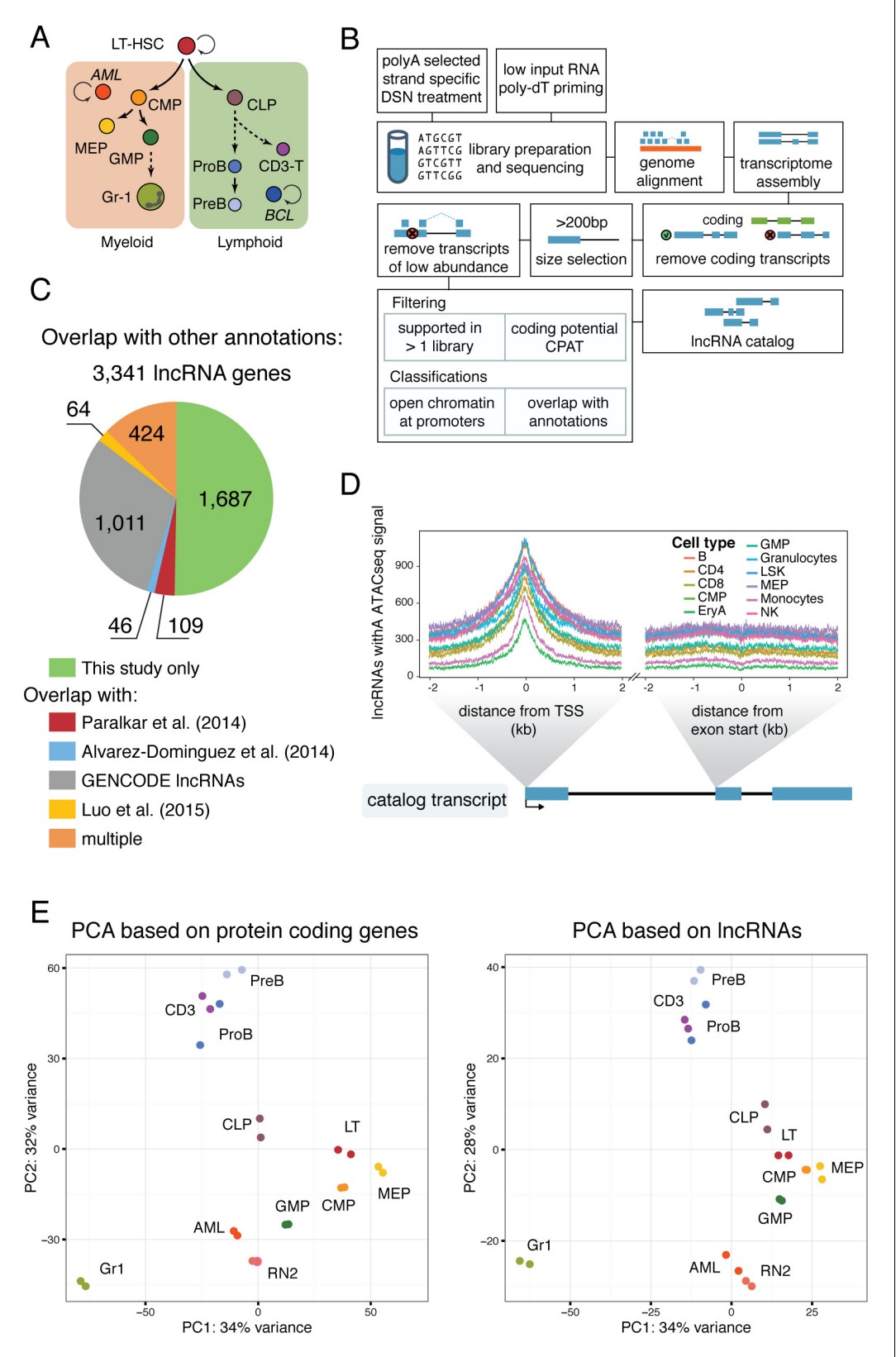

**Figure 1.** A lncRNA catalog of the murine hematopoietic system. (**A**) Schematic of analyzed cell types. Sorting plots can be found in *Figure 1—figure supplement 2*. (**B**) Pipeline for assembling the lncRNA catalog. See Materials and methods for full description. (**C**) Overlap of this study's lncRNA catalog with previously published

*Figure 1 continued on next page*

*Figure 1 continued*

non-coding annotations. Overlap is based on coordinates annotation of lncRNA exons. (D) ATAC-seq signal intensity, as a proxy for open chromatin, around the aggregated TSSs of our lncRNA catalog or the aggregated start sites of the second exons for the cell types indicated (ATAC-seq data from [*Lara-Astiaso et al., 2014*]). (E) Principal component analysis based on the most variable 500 protein coding genes (left) or catalog lncRNA genes (right). Each dot represents a biological replicate.

DOI: https://doi.org/10.7554/eLife.25607.002

The following figure supplements are available for figure 1:

**Figure supplement 1.** Validation of catalog transcriptional start sites and sorted progenitor identity.
DOI: https://doi.org/10.7554/eLife.25607.003
**Figure supplement 2.** Isolation of primary hematopoietic cell types
DOI: https://doi.org/10.7554/eLife.25607.004

---

used in the previous report. Overall, we have produced a comprehensive catalog of lncRNAs in the hematopoietic system that can serve as a foundation for understanding non-coding RNA function in these very well characterized cell types.

## lncRNAs and coding genes show similar expression patterns across hematopoietic differentiation

To understand the dynamics of lncRNA expression during hematopoietic development, we performed expression module analysis based on our RNAseq datasets. We identified differentially expressed lncRNAs and protein-coding genes that showed the same expression patterns. The modules followed expected groupings, with enriched expression in either myeloid, lymphoid, or progenitor compartments. When representing the 15% most variable lncRNAs within each module and the same number of coding genes, we identify many genes that are well-established drivers of hematopoietic differentiation and progenitor maintenance (*Figure 2A*). Among the genes with enriched expression in LT-HSCs, we noted the *MDS1* and *EVI1* complex locus (*Mecom*). These are known to regulate hematopoietic stem cell self-renewal (*Yuasa et al., 2005*). *Hoxa9* and *Meis1*, landmarks of MLL-AF9 AML self renewal (*Krivtsov et al., 2006*), are found in the module corresponding to genes enriched in both progenitors and our AML samples. Key regulators in lymphoid development such as *Rag1/2*, *Ebf1* and *Cd38* appear in the lymphoid-enriched module, while *Csf3r* and *Itgam*, also known as *CD11b*, part of the Mac-1 receptor, are in the module enriched for *Gr1* expression (*Figure 2A*). These expression patterns are therefore consistent with the published literature and underscore the robustness of our data.

We wondered whether these coordinated lncRNA-gene expression patterns were a consequence of RNAs being produced from a bidirectional promoter leading to divergent lncRNA transcripts. Expression correlation of lncRNAs with a divergent transcript has been reported in embryonic stem cell differentiation (*Dinger et al., 2008*; *Sigova et al., 2013*) and human B and T cell lineages (*Casero et al., 2015*). A general model has even been proposed, whereby divergent lncRNAs regulate the expression of the associated coding gene during differentiation (*Luo et al., 2016*). When we examined expression levels across cell types between lncRNAs and their closest gene neighbors, we indeed detected some level of correlation (*Figure 2B*). However, this correlation was not exclusive to divergent transcripts, as a similar level of correlation was observed for other genomic organizations (*Figure 2B*). In the AML datasets, we observed enriched binding of transcription factors known to play a role in maintaining the transcriptional landscape of this model of leukemia (*Roe et al., 2015*) around the TSS of lncRNAs in our catalog (*Figure 2—figure supplement 1*). This suggests that lncRNA expression through development and disease is regulated by the same mechanisms as coding genes, hence leading to generally similar expression patterns.

## lncRNAs are specific to distinct hematopoietic cell types and to AML

Our gene co-expression analysis highlighted the existence of lncRNAs that are expressed in the same cell types, and with the same level of specificity, as the known master regulators of hematopoietic development. This raised the hypothesis that some of these lncRNAs, whose expression is tightly regulated during differentiation, could be key regulators of cell fate choice. To explore this possibility, we performed differential expression analysis and identified lncRNAs that were enriched

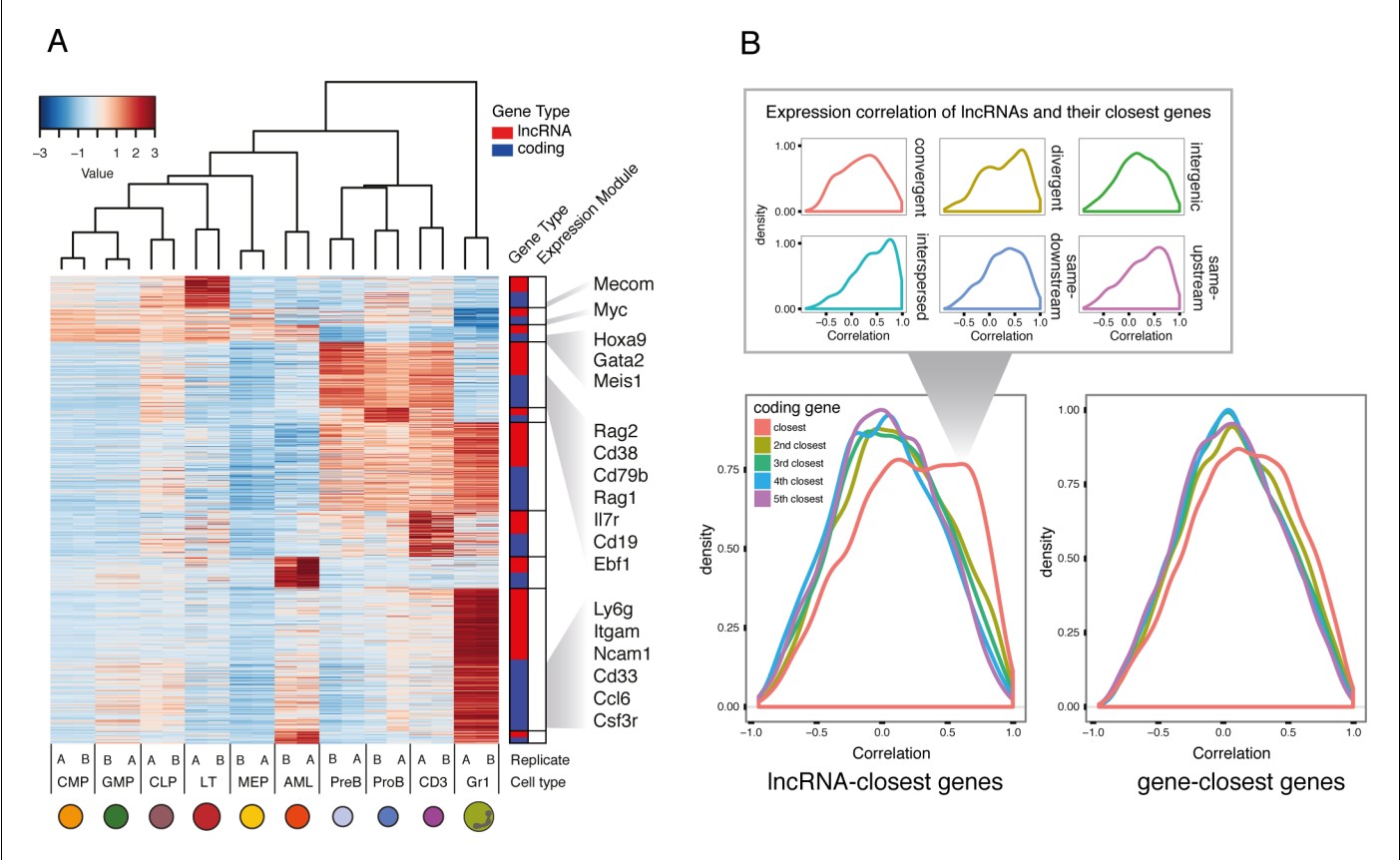

**Figure 2.** Expression profiles of lncRNAs through hematopoietic differentiation resemble broadly the profiles of protein-coding. (**A**) Expression data heatmap of the 15% most variable lncRNAs in each co-expression cluster, and the same number of most variable coding genes. Key hematopoietic genes are highlighted. (**B**) Density plot of correlation of expression for lncRNAs and their fifth closest coding genes. The density of the closest gene is categorized by genomic organization (right). As a comparison, correlation of genes and their closest genes is shown. Interspersed lncRNAs are not included in the main plot.

DOI: https://doi.org/10.7554/eLife.25607.005

The following figure supplement is available for figure 2:

**Figure supplement 1.** Regulatory landscape at the TSS of coding genes and lncRNAs.

DOI: https://doi.org/10.7554/eLife.25607.006

in hematopoietic stem cells, shared by the progenitor populations while showing lower expression in differentiated cell types, or enriched exclusively in the lymphoid compartment (*Figure 3A*, *Supplementary file 2*). This produced a list of candidates that could potentially function during self-renewal or differentiation (*Supplementary file 2*).

We also noticed many lncRNAs with enriched expression in AML, as well as shared expression between AML and other cell types. In our efforts to identify lncRNAs that are functionally relevant in the hematopoietic system, we focused on this AML model, given its ease of manipulation in vitro and in vivo and the availability of rapid in vitro and in vivo phenotypic assays. We selected a set of lncRNAs with varying levels of expression and a range of expression patterns for a pilot shRNA screen to test the effects of lncRNA depletion in a transplantable model of MLL-AF9/NRAS$^{G12D}$ AML (*Figure 3B*).

## An in vivo shRNA screen identifies lncRNAs required for leukemia development

For our screen, we selected a set of 120 lncRNAs that spanned the entire range of expression levels and included a diversity of expression patterns. For example, we chose lncRNAs that were AML

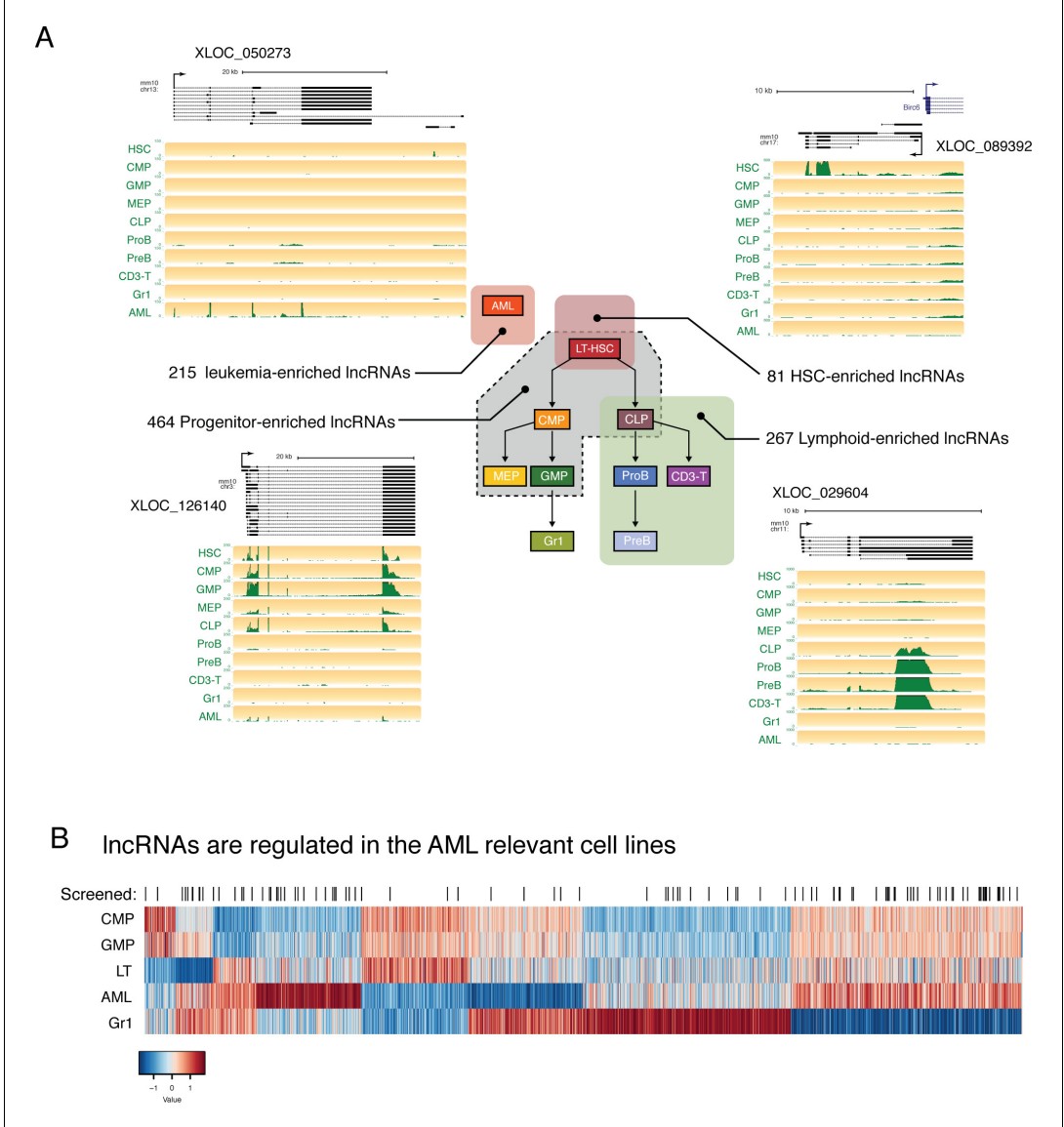

**Figure 3.** lncRNAs are differentially regulated through hematopoietic differentiation. (**A**) Genome Browser plots showing RNAseq coverage for representative lncRNA gene models in each of the indicated expression profiles along with the corresponding number of differentially expressed genes. (**B**) Heatmap of expression for lncRNAs differentially regulated between AML and relevant myeloid cell types. lncRNAs included in the in vivo AML screen are highlighted by black lines.

DOI: https://doi.org/10.7554/eLife.25607.007

specific or shared between AML and progenitors, as well as a variety of other patterns (*Figure 3B*). We also chose lncRNAs with different relative expression levels ranging from abundant to lowly expressed. We used the shERWOOD algorithm (*Knott et al., 2014*) to predict highly potent shRNAs targeting each lncRNA candidate. Because of the high isoform complexity in our assembled lncRNA catalog, a common characteristic of lncRNA assemblies, we could not simply predict on each individual transcript model assembled by our pipeline. We reasoned that targeting the regions of highest RNAseq coverage would maximize our chances of silencing the most abundant isoforms for each candidate. We also wanted the shRNA resource that we built to be applicable for studies beyond AML. Given the cell-type specificity observed in our data, we therefore decided to combine all reads for each lncRNA across libraries prior to coverage calculations, so as to focus our predictions on the most highly included exons.

We designed, cloned and sequence verified a library containing at least four hairpins per lncRNA into a doxycycline-inducible retroviral vector. As controls, we included hairpins against Renilla luciferase and Replication Protein A3 (*Rpa3*). MLL-AF9/NRAS$^{G12D}$ AML cells were infected at low multiplicity to minimize the probability of double infection, and were Neomycin-selected to eliminate non-infected cells. Infected AML cells were transplanted into mice, and hairpin expression was subsequently induced. To ensure a good representation of every hairpin during the 14 days that the cells proliferate in vivo, we performed virus production, infections and injections using pools of 50 shRNAs (*Figure 4B*). This number was based on previous experiments where cells were infected with a retrovirus carrying a neutral random nucleotide sequence (barcode), and the same experimental set up was followed to quantify representation of individual barcodes in tumors arising from populations infected with pools of different complexities (data not shown). shRNA representation was determined by high-throughput sequencing of hairpins amplified from genomic DNA extracted from the pre-injection pools and bone marrow samples taken 14 days post engraftment.

As expected, most shRNAs for *Rpa3* were depleted by the final time point. We did find an outlier, most likely a result of transcriptional silencing of the *Rpa3* hairpin, one of the known caveats of this sequencing-based readout. Importantly, most hairpins targeting Renilla luciferase, which is not expressed in the MLL-AF9 leukemia cells and serves as a negative control, were not significantly changed during the 2-week time course. In order for a particular lncRNA to be selected for more detailed follow-up, we required at least two hairpins to be significantly depleted at day 14 as compared to day 0 (FDR < 0.05) or one hairpin significantly depleted and a second hairpin depleted more than twofold (*Figure 4C*). This produced a list of 20 primary hits that were potentially required for leukemia maintentenance in vivo. These lncRNA candidates were prioritized for further study in an MLL-AF9/NRAS$^{G12D}$ AML cell-culture model.

To validate our in vivo screen and to assess whether AML cells were also dependent on these lncRNAs in cell culture, we performed competitive proliferation assays for all primary screen hits with two independent hairpins using a constitutive vector (*Figure 4E*). The shRNA-containing cells also expressed a green fluorescent protein (zsGreen), which we tracked over time to determine the ability of lncRNA-depleted cells to proliferate as compared to their uninfected counterparts. 14 of our 20 lncRNA candidates showed a depletion of over 50% for both hairpins over the 14-day time course, which represents a 70% validation rate of the initial hits (*Figure 4E*). Both our primary and validated hits displayed a range of different expression levels (*Figure 4D*). We selected 9 out of these 14 lncRNAs for follow up experiments based on the severity of the proliferation phenotype and its consistency across independent knockdowns (*Figure 4E*). We also included lncRNAs in a variety of arrangements with respect to their neighboring protein coding genes.

## lncRNAs are required for leukemia proliferation in vivo and in vitro

In order to study the effects of lncRNA knockdown, both at the level of global gene expression and at the level of protein abundance, we required a population of lncRNA-depleted cells with relatively homogeneous knock-down properties. To circumvent the proliferation defect caused by lncRNA knock down, we subcloned our constructs into an inducible retroviral vector, where hairpin expression was under the control of the TRE3G dox-inducible promoter. We then isolated two clonal lines for each hairpin to avoid any phenotypes linked to specific integration sites.

We assessed knockdown efficiency after 2 days of doxycycline induction by testing each sample with two independent primer pairs corresponding to each lncRNA. For four of our selected lncRNAs, we observed over 70% knockdown efficiency in at least one of the clonal lines for both primers and both hairpins. Additionally, two others had a more modest (~60%) but consistent knockdown across hairpins and primer pairs. We also examined the relationships between lncRNAs and their closest genes, with a particular interest in whether down-regulation of the lncRNA would lead to any expression changes. We did not observe any consistent trend, with most lncRNA knockdowns failing to induce any reproducible change of expression in either of their two flanking coding genes (*Figure 5A*).

Two of our candidate lncRNAs are divergently transcribed with respect to their neighboring genes (*lncRNA_041249* and *lncRNA_097790*) and one lncRNA annotation overlaps, in the antisense direction, with its closest coding neighbor (*lnc_166788*). We looked particularly at whether these might function by controlling the level of expression of their linked divergent coding transcript, as previously reported for ES cell differentiation (*Luo et al., 2016*). Knockdown of the two divergent

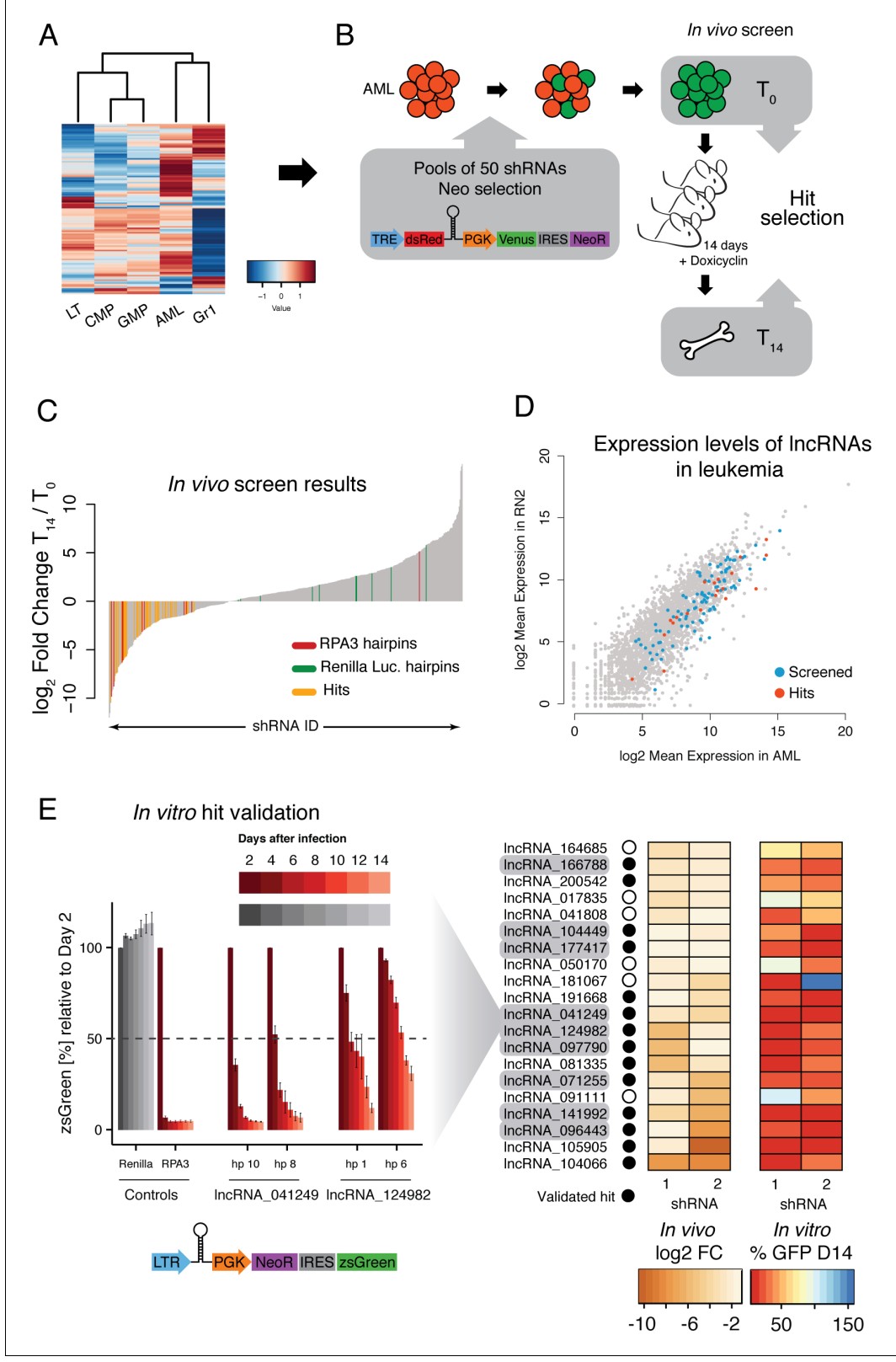

**Figure 4.** An in vivo shRNA screen identifies lncRNAs required for leukemia progression. (**A**) Expression profile for the lncRNAs included in the screen. (**B**) Screen outline: murine MLL-AF9/NRAS$^{G12D}$ AML cells were infected at low MOI with retrovirus-encoded pools of 50 shRNAs against candidate lncRNAs. Cells were selected to 100% infection and injected into sub-lethally irradiated mice (3–4 mice per pool, 1 million cells per mice). Hairpin

*Figure 4 continued on next page*

*Figure 4 continued*

abundance was estimated by high-throughput sequencing of genomic DNA from the initial injection pools and from whole bone marrow 14 days post-injection. (C) Fold change between final and injection time points for each hairpin in the screen. Renilla luciferase and *Rpa3* hairpins are highlighted in green and red, respectively. Hairpins of lncRNAs identified as hits are highlighted in yellow. (D) Scatterplot of mean lncRNA expression across two biological replicates in sorted bone marrow AML cells and in vitro culture AML cells. Screened lncRNA and hits are highlighted in blue and red, respectively. (E) One-by-one in vitro competition assay. AML cells were infected with a retrovirus constitutively expressing a single shRNA targeting the indicated lncRNA candidate. The percentage of zsGreen-positive cells relative to day 2 post-infection was followed over the course of 14 days. Bar graphs for controls and two representative lncRNAs (with two hairpins per lncRNA) are shown on the left. Values are the average of four biological replicates; error bars show s.e.m. A summary of the in vivo fold change and the percentage of zsGreen at the latest time point in vitro for all lncRNA hits are shown on the right displayed as a heat map.

DOI: https://doi.org/10.7554/eLife.25607.008

The following figure supplement is available for figure 4:

**Figure supplement 1.** Competitive proliferation assays for all the lncRNAs identified as hits in the in vivo screen, with two hairpins per lncRNA.

DOI: https://doi.org/10.7554/eLife.25607.009

---

lncRNAs, *lncRNA_041249* and *lncRNA_097790*, did not cause any changes in the divergent coding transcripts, *Srsf5* and *Rmb27*, respectively (corresponding to 'Upstream gene' in *Figure 5A*). The only case where we found depletion of a lncRNA to affect its closest neighboring gene was *lnc_166788*. This lncRNA is predicted to overlap with a *Gata2* isoform in a head-to-head configuration, although our libraries did not show much support for such an overlapping configuration. Our data indicate that this lncRNA might be involved in promoting *Gata2* expression and hence its depletion reduces *Gata2* mRNA levels. Although this is a very attractive possibility given the key role Gata2 plays in hematopoietic progenitor maintenance, further experimentation will be required to understand the precise nature of any such interaction.

## lncRNAs promote leukemia survival by maintaining leukemia stem cell signatures

We next aimed to understand how lncRNA expression promoted tumorigenesis in AML. Gene expression and immunophenotyping revealed that three of the lncRNAs that scored in our screen promoted a leukemic stem cell state, since depletion of any of these activated a default myeloid differentiation program. Gene Set Enrichment Analysis (GSEA) showed an enrichment for Macrophage Differentiation genes (Ingenuity Pathway Analysis list) upon knockdown of *lnc_071255* (already annotated as Pvt1), *lnc_104449*, or *lnc_177417*, while leukemia stem cell signatures (*Krivtsov et al., 2006*) showed enrichment in the control knockdown (*Figure 5B*). Additionally, depletion of these same lncRNAs led to the upregulation of the myeloid differentiation cell surface marker CD11B (also known as Mac-1) and downregulation of the stem cell marker c-Kit, as shown by flow cytometry (*Figure 5C*). This resembles the phenotype observed upon withdrawal of the oncogenic MLL-AF9 fusion protein (*Zuber et al., 2011b*) and suggests at least some degree of leukemia-specific dependency for these three lncRNAs.

To ask whether the nine lncRNAs we identified to be required for AML could be playing a general role in other highly proliferative cell types, we used the murine breast cancer cell line, 4T1, in a the same cell culture competition assay we used for AML hit validation (*Figure 4E*, *Figure 5—figure supplement 1*). While *Rpa3* was required for proliferation of 4T1 cells in vitro, none of our lncRNAs showed depletion even after 14 days in culture (*Figure 5—figure supplement 1*), indicating that these lncRNAs are not required for general cell or tumor survival and growth, but rather are specifically required for leukemia maintenance.

One of the main drivers of AML proliferation is the proto-oncogene, Myc. We therefore examined whether Myc levels were affected by lncRNA depletion in our AML model. Indeed, we observed a strong reduction of Myc protein levels upon *lnc_071255/Pvt1*, *lnc_104449*, or *lnc_177417* knockdown. Intriguingly, the mRNA levels of *Myc* were not reduced in a comparable manner to the protein levels for the *lnc_071255/Pvt1* and *lnc_104449* as compared to *lnc_177417*. Due to their

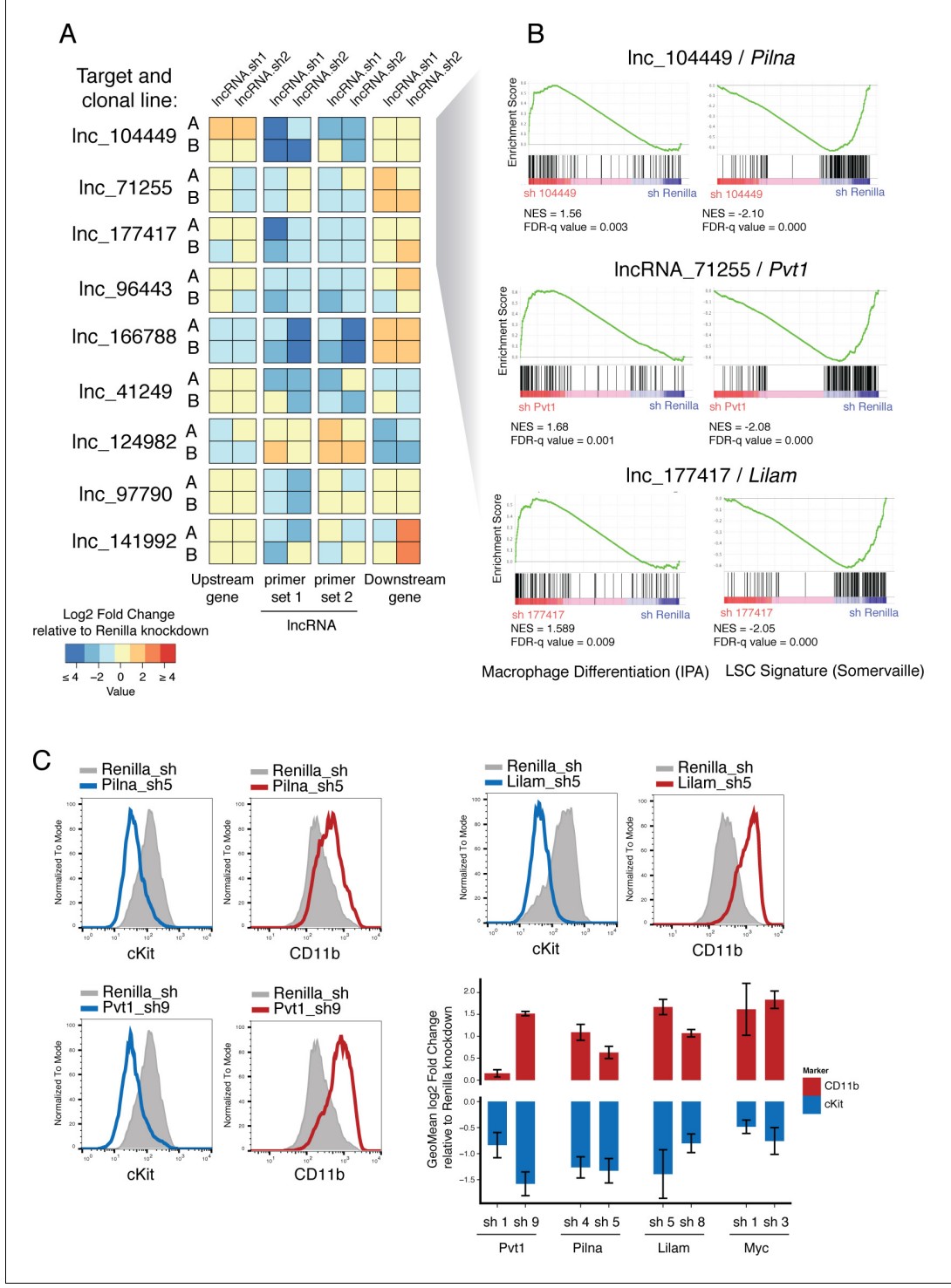

**Figure 5.** Characterization of top validated candidate lncRNAs. (**A**) Summary qPCR data shown as log2 fold change expression relative to Renilla Luciferase knockdown. Average values shown as a heatmap from three biological replicates (independent dox inductions) and two independent clonal cell lines. Relative expression was calculated as delta delta Cq using *Gapdh* as reference. For each lncRNA knockdown (rows), relative expression for the lncRNA itself, and its upstream and downstream genes are shown (columns). (**B**) Enrichment for Macrophage Differentiation (Ingenuity Pathway Analysis gene set) and Leukemia Stem cell Signature (***Krivtsov et al., 2006***) gene sets upon lncRNA knockdown was performed using Gene Set Enrichment Analysis. (**C**) Flow cytometry analysis of cKit and CD11b (Mac-1) surface markers upon lncRNA or Renilla knockdown. A representative plot per

*Figure 5 continued on next page*

*Figure 5 continued*

lncRNA is shown on the left. Geometric Mean fold change relative to Renilla knockdown is shown as the average of four biological replicates. Error bars represent s.e.m.

DOI: https://doi.org/10.7554/eLife.25607.010

The following figure supplement is available for figure 5:

**Figure supplement 1.** Validated lncRNA hits are not generally required for proliferation.

DOI: https://doi.org/10.7554/eLife.25607.011

expression and genomic position, we named *lnc_177417 Lilam* (leukemia-induced LncRNAaffecting Myc) and *lnc_10449 Pilna* (progenitor-induced lncRNA neighboring Ak3). *lnc_071255*, which corresponds to the previously annotated lncRNA *Pvt1*, was recently shown to correlate with MYC protein, but not *Myc* mRNA levels. In human breast cancer, *Pvt1* has been proposed to act by stabilizing MYC protein (*Tseng et al., 2014*). We also examined whether MYC target genes were affected by depletion of our lncRNA candidates using GSEA and found that knockdown of any of the lncRNAs that induced a myeloid differentiation phenotype resulted in a decrease in a Myc target gene-expression signature (*Figure 6—figure supplement 1*).

To understand whether MYC and its target genes were mediating the proliferation and differentiation phenotypes caused by reduction in lncRNA levels, we expressed the *Myc* gene in the context of lncRNA knockdown. Expression of *Myc* rescued the proliferation phenotypes for *Pvt1* and *Lilam* knockdown, as shown by a growth competition assay. To exclude a general growth effect, we specifically looked at the stem cell marker cKit and myeloid marker CD11b by immunophenotyping, as well as at the morphological phenotype by Giemsa staining after inducing knockdown with or without *Myc* co-expression for 48 hr. Both datasets were consistent with *Myc* expression rescuing the differentiation phenotype observed upon knockdown for each of these three lncRNAs (*Figure 6*, *Figure 6—figure supplement 1*). This strongly suggests an epistatic relationship between our lncRNAs and *Myc* and indicates that *Pvt1*, *Lilam* and *Pilna* exert their functions through Myc in this context.

We additionally examined the role of these lncRNAs in two other AML models (*Figure 6—figure supplement 2*). We observe depletion of all five tested lncRNAs (*Lilam*, *Pilna*, *Pvt1* plus two validated lncRNA hits with consistent knockdown) in the MLL/ENL model (similar to the MLL/AF9). In a different model, where Myc is one of the drivers, we observed a less striking phenotype. This is unsurprising since enforced Myc expression rescues the phenotype of *Lilam*, *Pilna* and *Pvt1* (*Figure 6—figure supplement 2*). This further supports an AML role for these lncRNAs.

## lncRNA *Pilna* is required for myeloid differentiation

The three lncRNAs, upon which we had focused, could exert effects specifically in tumors, or they could play broader roles, also affecting normal development. Their impact on Myc could suggest a more pervasive effect. However, previous studies have shown that chemical inhibition of Brd4, which also leads to Myc downregulation, has little or no effect for mouse hematopoiesis (*Zuber et al., 2011c*).

We performed competitive bone marrow transplantations with shRNA-expressing HSCs on lethally irradiated mice and monitored hematopoietic reconstitution via peripheral blood analysis. The shRNAs were sub-cloned into a lentiviral vector that also expressed the fluorescent protein, zsGreen, under the SFFV promoter, to ensure expression in the early progenitors and allow for identification of transduced cells. To distinguish donor from recipient cells, we took advantage of the syngeneic mouse strains CD45.2 and CD45.1. We used a combination of CD11b and Ly6G antibodies as a broad gate for all myeloid cells and we monitored the zsGreen percentage within the donor cells for this compartment (*Figure 7A*). If an shRNA targeted a gene that is required at some stage of myeloid differentiation, we would observe a reduction in the percentage of zsGreen-expressing cells.

We observed a significant depletion of zsGreen-expressing cells over time with either hairpin targeting *Pilna* as compared to Renilla knockdown, while little or no effect was observed for *Pvt1* and *Lilam* shRNAs (*Figure 7B*). When we examined the expression of these three lncRNAs, we observed that both *Pvt1* and *Lilam* were highly upregulated in leukemia cells (RN2) as compared to their most

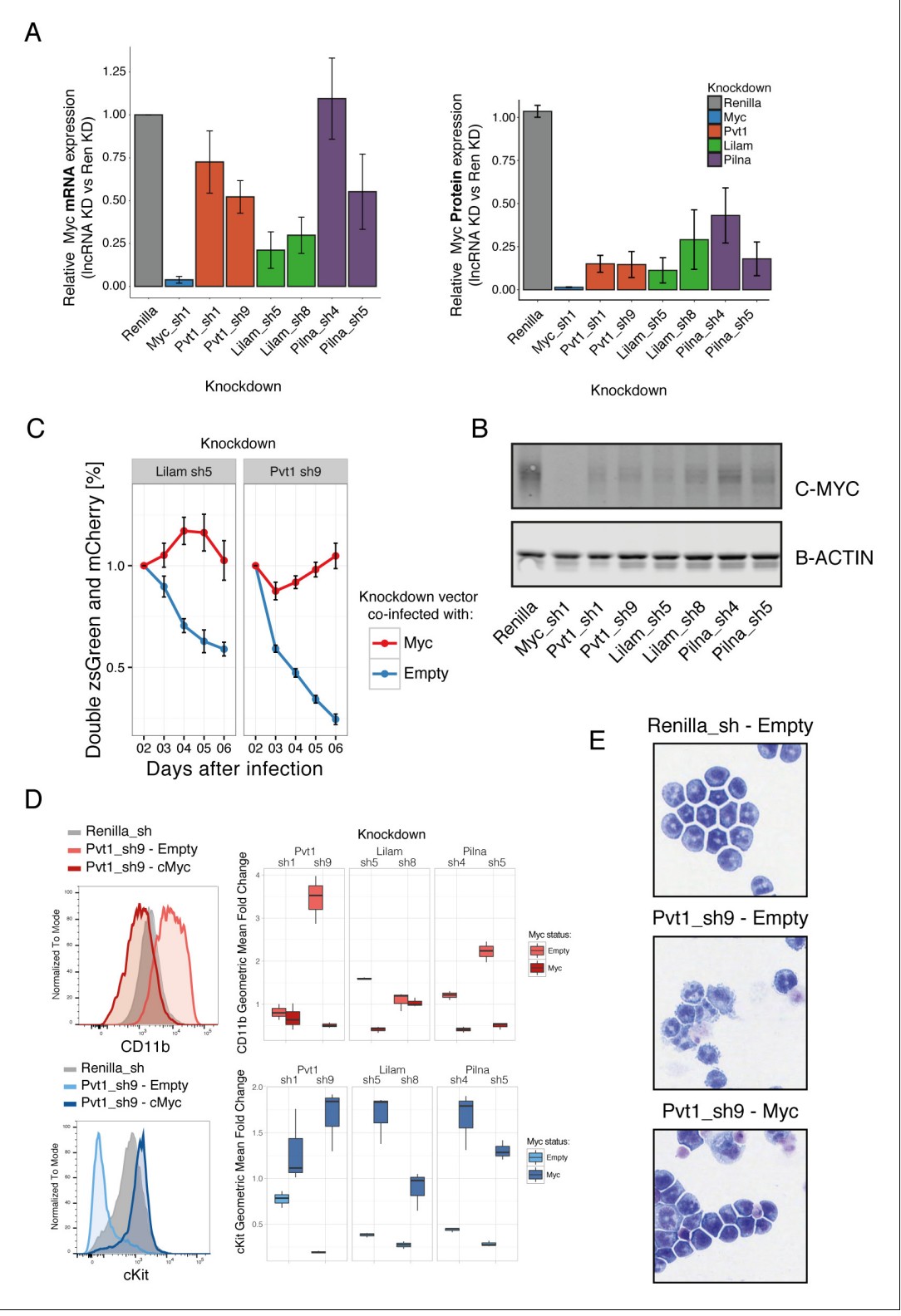

**Figure 6.** Three 'leukemia lncRNAs' affect MYC and enforced expression of this oncogene rescues proliferation and differentiation effects. (**A**) Relative *MYC* mRNA and MYC protein. Values represent the mean across three biological replicates, error bars represent s.e.m. (**B**) A representative western blot from (**A**). (**C**) Percentage of cells co-expressing either a constitutive shRNA vector (zsGreen) and a MSCV-based vector expressing mCherry and either Myc or empty. Data is relative to day 2 after infection. Average values across three replicates shown, error
*Figure 6 continued on next page*

*Figure 6 continued*

bars represent s.e.m. (**D**) Immunophenotyping of AML cells expressing either an inducible Renilla shRNA vector, co-expressing lncRNA Pvt1 knockdown and MYC or lncRNA knockdown without MYC (empty). On the right, quantification of the relative expression compared to Renilla is shown as fold change of the geometric mean for each fluorophore. Data obtained from three biological replicates. (**E**) Wright-Giemsa staining of cytospun cells expressing the indicated inducible shRNA vectors with or without MYC.

DOI: https://doi.org/10.7554/eLife.25607.012

The following figure supplements are available for figure 6:

**Figure supplement 1.** Myc target genes are affected by lncRNA knockdown and enforced Myc expression rescues the differentiation phenotype.

DOI: https://doi.org/10.7554/eLife.25607.013

**Figure supplement 2.** lncRNA requirements in other AML models.

DOI: https://doi.org/10.7554/eLife.25607.014

---

similar normal progenitors, GMPs, or HSCs (*Figure 7C*). On the other hand, *Pilna* is broadly expressed across hematopoietic progenitors. Our results suggest a more general role in hematopoiesis for *Pilna*, while lncRNAs that show leukemia-enriched expression, *Pvt1* and *Lilam,* could be dispensable for myeloid differentiation. This resource and the defined cohorts of leukemia-enriched lncRNAs could be further used to identify other lncRNA with potential leukemia-specific roles.

## Discussion

Long non-coding RNAs are emerging as a recently recognized class of regulators that have broad impact in biology. To enable the study of this impact and breadth of function in a well-characterized developmental model, we have produced a de novo catalog of lncRNAs from a representative sampling of cell types throughout the hematopoietic lineage and from two models of hematological malignancies. To globally characterize the likely impact of lncRNAs in normal and malignant hematopoiesis, we chose a subset of RNAs with a range of expression levels and patterns and tested whether these had a function in a murine model of AML. We performed a loss-of-function screen of 120 candidates and found that 20 of these were required for in vivo leukemia progression. We further characterized nine lncRNAs, all of which were required for leukemia cell proliferation in vitro but were dispensable in at least one other unrelated cancer model. A subset of these lncRNAs functioned in maintaining leukemia stem cell signatures; the leukemic blasts acquired a myeloid differentiated phenotype upon lncRNA knockdown. Amongst such lncRNAs was *Pvt1*, which has been shown previously to act via the transcription factor MYC (*Tseng et al., 2014*). This also appeared true in our model, as enforced expression of MYC rescued the proliferation and differentiation phenotypes observed upon depletion of *Pvt1*. Although the same phenotype, including Myc-mediated rescue, was also observed for two other lncRNAs (*Lilam* and *Pilna*) in leukemia, only *Pilna* showed an effect in normal reconstitution of the myeloid lineage. This is consistent with the progenitor-wide expression of *Pilna* and suggests a general role in hematopoiesis for this lncRNA. Expression of lncRNAs *Pvt1* and *Lilam* is highly induced in the leukemia context and they could, on the other hand, be dispensable for normal myelogenesis.

### Regulation of lncRNA expression

Expression analysis of our annotated lncRNAs revealed that differential regulation between cell types and at different stages of differentiation. Much attention has been focused on noncoding RNAs transcribed in the opposite orientation from the same promoter region as coding transcripts, known as divergent transcripts. This raises questions regarding whether these noncoding transcripts are truly functional, or whether they are simply a byproduct of transcription from an activated promoter. Expression of divergent lncRNAs has been shown to correlate with their corresponding protein-coding counterparts, and even to regulate their counterparts in pluripotent stem cells (*Casero et al., 2015*; *Dinger et al., 2008*; *Luo et al., 2016*; *Sigova et al., 2013*).

We indeed find a correlation between lncRNA expression and that of the closest protein-coding gene; however, this is not an exclusive property of divergent transcripts. We also observe correlation for other lncRNA-gene genomic orientations, as well as for gene-gene pairs. We therefore

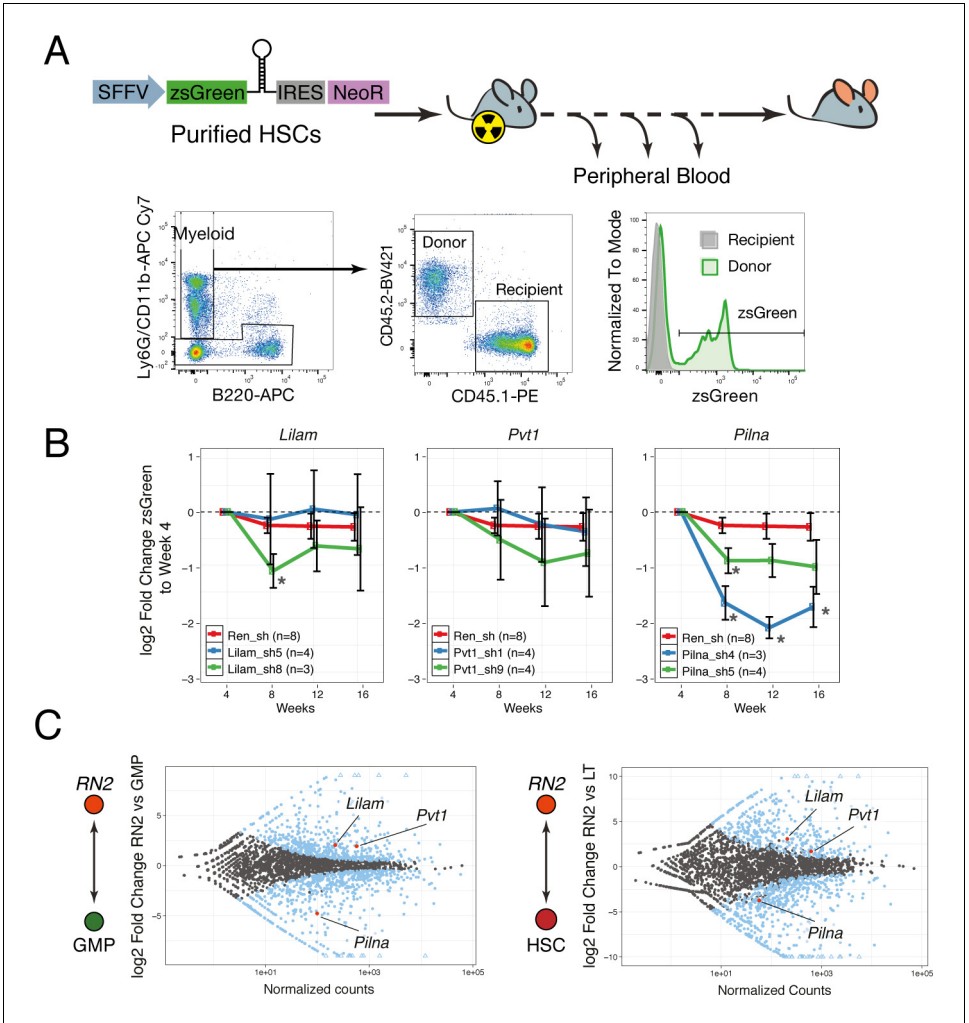

**Figure 7.** lncRNAs Pvt1 and Lilam are miss-regulated in leukemia while Pilna is required for myeloid reconstitution. (**A**) Outline of the peripheral blood analysis following bone marrow transplantation. (**B**) Relative zsGreen percentage to week 4 values. Values of zsGreen represent percentage within donor (CD45.2 +) for the myeloid compartment. Average log2 fold change shown, error bars represent s.e.m. Statistical significance was calculated using the Mann-Whitney/Wilcoxon Rank Sum Test (* p-value<0.05). (**C**) Relative expression of lncRNAs between leukemia cells (RN2) and two normal progenitor cells types, GMP (left) and LT-HSC (right) versus normalized read counts. Log2 fold change and normalized read counts obtained using DESeq2. Blue dots depict differentially expressed lncRNAs (FDR < 0.05), lncRNAs of interest are highlighted in red.

DOI: https://doi.org/10.7554/eLife.25607.015

hypothesize that this correlation in expression arises from shared regulatory mechanisms or common regulatory environments. For the two divergent lncRNAs amongst the ones we characterized, we did not find evidence for *cis* regulation of their corresponding divergent gene.

## lncRNA dependencies in AML

We probed the functional relevance of lncRNAs using AML as a model. By performing an shRNA screen targeting 120 lncRNAs, we identified 20 lncRNAs whose expression confers a proliferative advantage to leukemia cells in vivo. This was striking particularly because we did not solely target AML-enriched lncRNAs but rather cast our net broadly. Thus, our findings emphasize the general importance for lncRNAs in leukemia and possibly normal hematopoiesis. We tested whether the lncRNAs that impacted AML were also required for proliferation of the breast cancer cell line 4T1. In this model, we observed no significant effect of knockdown. This could be due to cell type-restricted

expression or function but can be taken as support for the specificity of the effects observed in AML.

Three lncRNAs emerged as candidates required for leukemia progression, as their depletion led to leukemic blast differentiation. Amongst these was *Pvt1*, a lncRNA previously reported to be a marker of poor prognosis in colorectal cancer (*Takahashi et al., 2014*) and shown to be required in MYC-driven cancers (*Tseng et al., 2014*). The mechanism for human *PVT1* function, as shown in breast cancer, involves *PVT1* stabilization of the MYC protein. We observe a similar effect in our leukemia model, with MYC protein levels being disproportionately reduced when *PVT1* is depleted.

Enforced expression Myc rescued the effects of *Pvt1*, as well as *Lilam* and *Pilna*, knockdown in MLL-AF9 leukemia cells. This suggests that these lncRNAs have a direct or indirect functional relationship with Myc, an oncogene upon which this AML model heavily depends. This finding adds a layer to the already complex regulatory landscape of AML, and fits in with published results that have shown similar phenotypes when Myc levels are reduced by disturbing bromodomain containing 4 (Brd4) (*Zuber et al., 2011c*) or the Myc superhancer (*Shi et al., 2013*). lncRNA *Pilna*, but not *Pvt1* and *Lilam* are required for the myeloid lineage during bone marrow reconstitution. This is consistent with the pan-progenitor expression of this lncRNA in contrast with the AML-specific nature of *Pvt1* and *Lilam*. Further studies will be required to determine the breadth of lncRNA roles in the hematopoietic system, though this gives us reason to believe that the normal setting, just like AML, will be heavily dependent on these non-coding transcripts.

Considered as a whole, we have produced an extensive resource of lncRNA expression in the hematopoietic system and demonstrated that a substantial percentage of these are functional, at least in a model of malignant development. One of these lncRNAs, expressed across most hematopoietic progenitors, is additionally required in myeloid reconstitution. The data and tools that we have produced should serve a useful purpose in promoting studies in both the normal and leukemic context.

## Materials and methods

### Animal work

Tissue extraction for transcriptome studies and the in vivo shRNA screen were performed in Cold Spring Harbor Laboratories (NY, USA). RNAseq libraries for transcriptomic assembly and differential expression were prepared from female C57BL/6 (6–12 weeks old) purchased from Charles River (Wilmington MA). For the in vivo shRNA screen, female B6.SJL-Ptprc^a/BoAiTac (also known as LY5.1, 8–12 weeks old) were purchased from Taconic (Hudson, NY). All these experiments were approved by the Cold Spring Harbor Animal Care and Use Committee.

Bone marrow transplantations of modified HSCs and subsequent analysis of peripheral blood from these mice were performed at the Cancer Research UK Cambridge Institute (Cambridge, UK). C57BL6J (6–12 weeks old) were purchased from Charles River (Kent, England). C57BL/6-LY5.1 females were purchased from Charles River (Kent, England) and used at 9–12 weeks old. These animal procedures were conducted in accordance with project and personal licenses issued under the United Kingdom Animals (Scientific Procedures) Act, 1986.

### Low input RNAseq library preparation for transcriptome assembly and differential expression

Low input libraries were prepared from C57BL6 mice for all normal hematopoietic cell types depicted in *Figure 1A*. Femurs and tibias were flushed with Hanks Balanced Salt Solution (Gibco) supplemented with 1% heat inactivated Fetal Bovine Serum (HyClone). For lineage depletion, the Mouse Lineage depletion kit (Milteny Biotec 130-090-858, Germany) was used. To preserve RNA integrity, the procedure was carried out at 4°C. Cells were sorted following previously published gating strategies, shown in *Figure 1—figure supplement 2* and RNA was extracted using NucleoSpin RNA XS (Machery Nagel, Bethlehem, PA), including DNase treatment. Between 3000 and 10,000 cells were used as library input using Clontech's SMARTer Ultra Low input RNA Sequencing Kit (Cat. No. 634823, Clontech, Mountain View, CA). cDNA was sheared to 300 bp on a Covaris LE220 and used as input for Clontech's low input library preparation for Illumina Sequencing kit (Cat. no. 634947). Libraries were sequenced paired-end 100 bp on an Illumina HiSeq2500. Due to the low

percentage that HSCs represent in the bone marrow, the libraries were prepared form low numbers of cells as stated above and two biological replicates were produced for each library.

## RNAseq library preparation for transcriptome assembly

RNAseq libraries enriched for low-abundance transcripts were prepared for AML cell lines, lymphomas, and FACS-sorted PreB and ProB cells as follows. RNA was extracted using Trizol reagent (Thermo Fisher Scientific), DNase-treated, then polyadenylated transcripts were isolated using the Dynabead mRNA purification kit (Thermo Fisher Scientific). Purified mRNA was fragmented by heating to 98°C for 30 min in RNA storage buffer (Ambion), then converted to cDNA with random primers using the Superscript III RT kit (Invitrogen). Second-strand synthesis was performed in the presence of dUTP, then Illumina adapters were ligated onto the dsDNA fragments. To preserve strand identity, the uracil-containing cDNA strand was digested using USER enzyme (NEB), then cDNA was amplified using adapter-specific PCR primers and purified using Agencourt AMPure XP beads (Beckman Coulter). The samples were subsequently enriched for low-abundance transcripts by Duplex-Specific Nuclease (DSN) treatment (Illumina) followed by low-cycle PCR, then gel-purified and cleaned up using Agencourt AMPure XP beads (Beckman Coulter, Brea, CA). Libraries were sequenced by paired-end 76 bp on an Illumina HiSeq2500.

## RNASeq data processing and catalog assembly

RNAseq libraries were mapped with STAR aligner (*Dobin et al., 2013*) against the mm10 mouse genome assembly using default parameters. Duplicate alignments were removed from the resulting BAM files with Picard (http://broadinstitute.github.io/picard). Transcriptome assembly was performed individually for each library with cufflinks (*Trapnell et al., 2010*) utilizing GENCODE Release M4 annotations. Individual transcriptome assemblies were merged with program cuffmerge (*Trapnell et al., 2010*). The resulting merged assembly was filtered by removing transcripts (a) consisting of a single exon or spanning fewer than 200 bp, (b) overlapping a coding exon in the same orientation, (c) having FPKM below 0.3, (d) having at least one exon supported by fewer than 40 reads in each library, (e) overlapping genes annotated as IG*, (f) having a coding probability estimated by CPAT below 0.5 (*Wang et al., 2013*). We also required that an intron-exon structure of a transcript was supported in at least two libraries. The intron-exon structure similarity of two transcripts was measured using Jacquard index of genomic intervals defined by their introns. A Jaccard index cutoff of 0.2 was used. In order to calculate the number of fragments mapping to each transcript in each library overlapping catalog genes were merged together. The fragment counting itself was performed with the program htseq-counts (*Anders et al., 2015*). These merged annotations were used for subsequent expression analyses.

## Cell culture maintanance

MLL-AF9;NRAS$^{G12D}$ AML cells were obtained from the Lowe laboratory (*Zuber et al., 2011a*) and cultured in RPMI-1640 with GlutaMax (Gibco), supplemented with 10% heat-inactivated FBS (HyClone) and 1% Penicillin/Streptomycin (Gibco) under 7.5% $CO_2$ culture conditions. The cell line established from this model is also known as RN2. MLL-ENL;NRAS$^{G12D}$ and Myc;NRAS$^{G12D}$;p53 null were kind gifts from Johannes Zuber (unpublished) and were kept in the same culture conditions as RN2. Platinum-A packaging (purchased from Cell Biolabs Inc, San Diego, CA) were cultured in DMEM containing 4.5 g/L glucose, 4 nM L-Glutamine and 110 mg/L Sodium Pyruvate, supplemented with 10% FBS (HyCLone) and 1% Penicillin/Streptomycin (Gibco) under 5% $CO_2$ culture conditions. 293 FT (purchased from Thermo Fisher Scientific) were cultured as per manufacturer's instructions. The mouse mammary tumour cell line 4T1 (purchased from ATCC, Manassas, VA) was cultured in DMEM high glucose (Life Technologies) supplemented with 5% fetal bovine serum (HyCLone), 5% fetal calf serum (HyCLone), non-essential amino acids (Life Technologies) and penicillin-streptomycin (Life Technologies). All cell lines tested negative for mycoplasma contamination RNA-capture ELISA.

## shRNA design and cloning

shRNAs were predicted using the shERWOOD computation algorithm (*Knott et al., 2014*). To select the best 4–5 shRNAs against each lncRNA with all its isoforms, we pooled the RNAseq data for the hematopoietic cell types and prioritized the regions of highest coverage. shRNAs were cloned into

the appropriate vectors, with ultramiR backbone: TRMPV-Neo (AML screen), ultramiR-zsGreen-NeoR (validation one-by-one knockdown), T3GRUMPV-Neo (clonal inducible cell lines) or ZIP-Neo (bone marrow transplantations), as previously described (*Knott et al., 2014*).

## Virus production

Virus production for was performed as previously described (*Wagenblast et al., 2015*). In brief, for MSCV-based retroviruses, Platinum-A packaging cells (Cell Biolabs) were plated on 10 cm dishes and transfected at ~70% confluency. A transfection mixture of with 20 µg of shRNA vector, 2.5 µg of VSV-G, 66.8 µl of 20 µm Pasha siRNA (Qiagen, Germantown, MD) and 62.5 µl 2M $CaCl_2$ was prepared and brought to 500 µl with $H_2O$. This mixture was vigorously bubbled into 500 µl of 2X HBS solution (50 mM HEPES, 280 mM NaCl, 1.5 mM $Na_2PO_4$, 12 mM Glucose,10 mM KCl) for 30–60 s and added to cells in 9 ml of supplemented DMEM. After 16 hr, media was changed to supplemented RPMI and then collected 24 hr, 36 hr and 48 hr after media change and filter through a 0.45-µm filter (EMD Millipore). When necessary, virus was concentrated using Retro-X concentrator (Clontech).

For third generation lentiviruses, virus was prepared in 15 cm dishes using 293FT cells (Thermo Fisher Scientific). The transfection mixture contained 32 µg of DNA vector, 12.5 µg of pMDL, 6.25 µg of CMV-Rev, 9 µg of VSV-G, 200 µg of Pasha siRNA, 125 2.5M of $CaCl_2$ brought to 1250 µl with $H_2O$ and bubbled into 1250 µl 2X HBS. Media was changed to IMDM supplemente with 10% heat-inactivated FBS right before transfection and collected in 16 ml of the same media. 38 ml of viral supernant was ultracentrifuged for 2.5 hr at 25,000 rpm at 4°C, and resuspended in 100 µl of D-PBS (Gibco).

## Pooled shRNA screening in vivo

Pools of 50 shRNA vectors were used to produce virus and transduce AML cells at a low multiplicity of infection to minimize double infections. Cells were treated with 500 µg $ml^{-1}$ G-418 (Roche Applied Science, Penzberg, Germany) from 2 days after infection until fully selected. $1 \times 10^6$ fully selected AML cells were injected in the tail vein of sublethally irradiated (4.5 Gy, 24 hr before injection) B6.SJL (CD45.1) female mice 6–8 weeks of age. For shRNA induction, animals were treated with doxycycline in the food (625 mg $kg^{-1}$, Harlan Laboratories/Envigo, South Easton, MA). Leukemic mice were euthanized 14 days after transplantation, at terminal disease stage, by $CO_2$. Cells were extracted from the bone marrow by flushing tibias and femurs and filtered (0.45 µm) to obtain a single cell suspension. For each pool, we required a minimum of three mice at the 14-day timepoint.

## shRNA library processing and analysis

Library preparation was performed as previously described (*Knott et al., 2014*). In brief, genomic DNA was extracted from the pre-injection pool and the bone marrow cell suspensions using the QIAamp Blood DNA Maxi Kit (Qiagen). For each sample, shRNA hairpins were extracted from genomic DNA in 96 separate 25-cycle PCR reactions where 2 µg of input DNA was included in each reaction. Following this initial PCR, Illumina adapters were added via PCR, and samples were processed on the Illumina MiSeq platform. Reads were extracted and mapped to the shRNAs of the corresponding pool using bowtie (allowing 0 mismatches). To analyze the depletion between the final timepoint and input, DESeq was used with Fit type 'local'. For follow up in vitro culture studies, we selected lncRNAs with at least two hairpins significantly depleted (FDR < 0.05) or one hairpin significantly depleted and another with at least twofold depletion.

## Cell culture competition assays

Cells cells were transduced with retroviral supernatant. Infection was assessed by flow cytometry analysis of a fluorescent reporter and kept to <30%. The percentage of cells expressing the fluorescent reporter over time was used to determine whether cells harboring the shRNA were being outcompeted by their uninfected counterparts.

## Immunophenotyping

RN2 cells harboring inducible shRNAs were cultured in complete media containing 1 μg/mL doxycy-cline (Clontech) for 4 days. Cells were stained in MACS Buffer (Miltenyi Biotech) with CD11b/Mac-1 PE-Cy7 and cKit APC for 30 min on ice and analyzed by flow cytometry. Plots were produced using FlowJo software and Geometric Mean of the appropriate channel was extracted for fold change calculations.

## Wright-Giemsa staining on cytospun cells

Cells were resuspended to 50,000 cells in 100 μl in MACS Buffer (Miltenyi Biotec). Of this buffer, 100 μl was first spun on the slides, followed by spinning of the cells for 5 min at 500 RPM. Slides were stained using the Kwik-Diff three step stain (Fixative; Eosin; Methylene Blue) from Thermo Fisher Scientific and imaged using the Aperio XT system at 40X.

## Relative mRNA quantification by RT-qPCR

Analysis of knockdown efficiency and relative gene expression changes was performed after induc-ing shRNA expression for 2 days. RNA was extracted using the RNeasy Mini Kit (Qiagen), including treatment with the DNase Set (Qiagen). Reverse transcription was performed using Superscript II (ThermoFisher Scientific), with 4 μg of RNA and 1 ul of 50 μM oligo(dT)$_{20}$. Primers were designed using IDT PrimerQuest tool or chosen from IDT's pre-designed set when available. Fast SYBR Green (ThermoFisher Scientific) was used for qPCR. Primer pair efficiency was assessed using serial dilutions of cDNA from untreated RN2 cells, and melting curves were examined to ensure the presence of only one amplicon. *Gapdh* was used as a housekeeping normalization control in the delta-delta-Ct analysis. For lnc_104449, we did not obtain a CT value upon knockdown with sh4 (sh1 in the heatmap) in 14 out of the 18 technical replicates after 40 cycles of qPCR. Therefore, we set those values to 40, if at all underestimating the actual knockdown with this hairpin.

## RNAseq from lncRNA-depleted AML and data analysis

Gene expression analysis was performed on RN2 cells harboring an inducible shRNA, treated with doxycycline for 2 days. RNA was extracted using TRIzol (Thermo Fisher Scientific). RNA sequencing libraries were prepared using TruSeq Stranded Total RNA Library Prep Kit (Illumina) and run on an Illumina HiSeq 2000. Reads were mapped to the mm10 genome assembly yielding at least $10^6$ aligned reads per sample. HTSeq-count (*Anders et al., 2015*) was used to calculate gene counts and subsequently input them into DESeq2 (*Love et al., 2014*) for quality control analysis, size nor-malization and variance dispersion corrections. Gene Set Enrichment Analysis (GSEA) was performed on variance-stabilized data.

## MYC western blotting

Whole cell lysates were prepared by resuspending cell pellets in SDS-PAGE loading buffer and run in a NuPAGE Novex 4–12% Bris-Tris Protein Gel. Transfer was performed using iBlot2.0 into PVDF membranes. Membranes were incubated with primary antibodies against MYC (Abcam AB32072 [Y69], UK) and b-actin (Abcam Ab6276) and secondary LiCor fluorescent antibodies. Membranes were imaged using the semi-quantitative LiCor Odyssey CLx system. MYC values are internally nor-malized to the B-ACTIN within each lane.

## Isolation, infection and short-term culture and transplantation of HSCs

Bone marrow from C57BL6 mice was extract by flushing, filtered through a 0.30 μm filter and lineage depleted (Mouse Lineage depletion kit, Milteny Biotec 130-090-858). Cells were stained with EPCR-PE, CD45-APC, CD150-PE/Cy7 and CD48-FITC. DAPI or LIVE/DEAD Fixable Violet Dead Cell Stain Kit (Thermo Scientific L34963) was used for dead cell exclusion. Sorting of highly pure E-SLAM HSC and sort-term (<24 hr) culture was performed as previously described (*Kent et al., 2009*). In short, 1000 alive EPCR +CD45+CD150+CD48- lineage negative cells were sorted in 100 μl of media: Iscove modified Dulbecco medium supplemented with 10 mg/mL bovine serum albumin, 10 μg/mL insulin, and 200 g/mL transferrin, 100 U/mL penicillin, 100 μg/mL streptomycin [purchased as BIT from StemCell Technologies], and $10^{-4}$ M β–mercaptoethanol (Gibco) plus 20 ng/mL interleukin-11 (IL-11; R and D Systems) and 300 ng/mL Steel factor (R&D Systems)). Ultracentrifuged viral

supernatant was added aiming at a final concentration of ~$2\times10^7$ IU/ml following the sorting. Each well was used to inject four animals; cells were washed prior to injecting to remove remaining viral particles.

## Peripheral blood extract and flow cytometry analysis

Blood was analyzed starting from 4 weeks after transplantation and every 4 weeks thereafter. 50–75 µl of blood was extracted from the animals' tail vein into heparin coated capillary tubes. Red blood cell lysis was performed using Ammonium Chloride Solution (Stem Cell Technologies). Samples were then stained with a pre-mix antibody cocktail and analysed in a LSR Fortessa (BD Biosciences). Flow data analysis was performed using FlowJo and statistical analysis using R Studio.

## Antibodies used for flow cytometry analysis and sorting

| Name | Antigen | Company | Clone |
|---|---|---|---|
| Sca1 PE | Ly-6A/E (Sca-1) | eBioscience | D7 |
| cKit APC | cKit/CD117 | eBioscience | 2B8 |
| CD150 PE-Cy7 | CD150 | Biolegend | TC15-12F12.2 |
| CD48 FITC | CD48 | Biolegend | HM48-1 |
| IL7Ra/CD127 PerCP-Cy5.5 | CD127 (IL7Ra) | eBioscience | A7R34 |
| cKit APC-Cy7 | cKit/CD117 | Biolegend | 2B8 |
| CD34 eFluor450 | CD34 | eBioscience | RAM34 |
| FgRg FITC | CD16/CD32 (FcgRII/III) | eBioscience | clone 93 |
| CD3 Pacific Blue | CD3 | Biolegend | 17A2 |
| Gr1 Alexa Fluor 700 | Ly-6G/Ly-6C (Gr-1) | Biolegend | RB6-8C5 |
| IgM eFluor450 | IgM | eBioscience | eB121-15F9 |
| B220 Alexa Fluor 700 | CD45R (B220) | Biolegend | RA3-6B2 |
| CD43 FITC | CD43 | eBioscience | eBioR2/60 |
| CD11b/Mac-1 PE-Cy7 | CD11b (Mac-1) | Biolegend | M1/70 |
| EPCR-PE | EPCR | Stem Cell Tech | RMEMPCR1560 |
| CD45-APC | CD45 | eBiosciences | 30-F11 |
| Ly6G-APC/Cy7 | Ly6G | eBiosciences | 1A8 |
| CD45.1-PE | CD45.1 | eBiosciences | A20 |
| CD45.2-BV421 | CD45.2 | eBiosciences | 104 |

## ATAC-seq data processing

ATAC-seq libraries (*Lara-Astiaso et al., 2014*) were first adaptor trimmed with trim-galore (http://www.bioinformatics.babraham.ac.uk/projects/trim_galore/). The resulting FASTQ files were mapped with bowtie2 (*Langmead and Salzberg, 2012*) aligner against mm10 mouse assembly using default parameters. Duplicate alignments were removed with Picard (http://broadinstitute.github.io/picard); alignments to mitochondrial chromosome and unplaced contigs were also removed. The coverages were calculated with bedtools (*Quinlan, 2014*).

## ChIP-seq data processing

ChIP-seq libraries (*Roe et al., 2015*) were mapped the same way as ATAC-seq. The heatmaps were generated with deeptools, using computeMatrix and plotHeatmap programs (*Ramírez et al., 2016*).

## Gene co-expression modules

The read counts were first transformed variance-stabilizing transformation in DESeq2 (*Love et al., 2014*). The modules themselves were obtained with WGCNA R package (*Langfelder and Horvath, 2008*). WGCNA analysis was performed using blockwiseModules function. Signed correlation

networks were used and the power for soft-thresholding was selected according to scale-free topology criterion. The minimum co-expression module size was set to 300 and merge cut parameter to 0.25.

### Differential expression analysis

Differential-expression analysis was performed with voom/limma (*Law et al., 2014*; *Ritchie et al., 2015*) utilizing empirical Bayes method (*Smyth, 2004*). Prior to differential-expression analysis, genes that did not attain counts-per-million of at least two in more than two libraries were filtered out.

### Correlation analysis

Spearman correlation was calculated for relevant pairs of genes using variance-stabilized expression values (*Love et al., 2014*) across libraries.

## Acknowledgements

The authors would like to thank the Flow Cytometry Core, the Biological Resource Unit, the Histopathology Core and the Research Instrumentation Core at Cancer Research UK Cambridge Institute for their support throughout this project. This work was also performed with assistance from CSHL Shared Resources, which are funded, in part, by the Cancer Center Support Grant 5P30CA045508. We would also like to thank Rebecca Berrens for assistance with high throughput qPCR assays, Osama El Demeresh for help consolidating sequencing data and Abigail Shea for assistance with some of the proliferation assays.

## Additional information

### Funding

| Funder | Grant reference number | Author |
|---|---|---|
| Boehringer Ingelheim Fonds | PhD Fellowship | M Joaquina Delás |
| "la Caixa" Foundation | Graduate Studies Fellowship | M Joaquina Delás |
| Damon Runyon Cancer Research Foundation | DRG-2016-12 | Leah R Sabin |
| National Institutes of Health | R01 HG007650 | Andrew D Smith |
| Cancer Research UK | | Gregory J Hannon |
| Howard Hughes Medical Institute | Investigator | Gregory J Hannon |
| Wellcome Trust | Investigator | Gregory J Hannon |
| Royal Society | Wolfson Professorship | Gregory J Hannon |

The funders had no role in study design, data collection and interpretation, or the decision to submit the work for publication.

### Author contributions

M Joaquina Delás, Conceptualization, Data curation, Formal analysis, Supervision, Validation, Investigation, Visualization, Methodology, Writing—original draft; Leah R Sabin, Conceptualization, Data curation, Formal analysis, Supervision, Investigation, Methodology, Writing—review and editing; Egor Dolzhenko, Data curation, Formal analysis, Validation, Writing—review and editing; Simon RV Knott, David R Kelley, Resources, Formal analysis; Ester Munera Maravilla, Data curation, Investigation; Benjamin T Jackson, Sophia A Wild, Eva Maria Stork, Formal analysis, Investigation; Tatjana Kovacevic, Investigation, Methodology; Meng Zhou, Software, Formal analysis, Investigation; Nicolas Erard, Resources, Data curation, Formal analysis, Visualization; Emily Lee, Data curation, Formal analysis, Investigation; Mareike Roth, Inês AM Barbosa, Johannes Zuber, Resources, Methodology; John

L Rinn, Resources, Formal analysis, Methodology, Writing—review and editing; Andrew D Smith, Resources, Formal analysis, Supervision, Writing—review and editing; Gregory J Hannon, Conceptualization, Supervision, Funding acquisition, Project administration, Writing—review and editing

### Author ORCIDs
M Joaquina Delás (iD) https://orcid.org/0000-0001-9727-9068
Eva Maria Stork (iD) http://orcid.org/0000-0002-9629-8950
Meng Zhou (iD) http://orcid.org/0000-0003-1487-5484
Johannes Zuber (iD) http://orcid.org/0000-0001-8810-6835
Gregory J Hannon (iD) http://orcid.org/0000-0003-4021-3898

### Ethics
Animal experimentation: For animal experiments conducted at Cold Spring Harbor Laboratory, all the animals were handled according to the approved institutional animal care and use committee (IACUC) protocol (#14-11-18). For animal experiments conducted at CRUK Cambridge Institute, all the animals were handled according to project and personal licenses issued under the United Kingdom Animals (Scientific Procedures) Act, 1986 (PPL 70/8391).

### Decision letter and Author response
Decision letter https://doi.org/10.7554/eLife.25607.030
Author response https://doi.org/10.7554/eLife.25607.031

## Additional files
### Supplementary files
• Supplementary file 1. lncRNA catalog
DOI: https://doi.org/10.7554/eLife.25607.017

• Supplementary file 2. Lists of lncRNAs in different expression groups from *Figure 3*
DOI: https://doi.org/10.7554/eLife.25607.018

• Supplementary file 3. lncRNA nomenclature in new versus old versions of the catalog (for reagent requests).
DOI: https://doi.org/10.7554/eLife.25607.019

• Supplementary file 4. Hairpins used in this study
DOI: https://doi.org/10.7554/eLife.25607.020

• Supplementary file 5. qRT-PCR primers used in this study
DOI: https://doi.org/10.7554/eLife.25607.021

• Transparent reporting form
DOI: https://doi.org/10.7554/eLife.25607.022

### Major datasets
The following dataset was generated:

| Author(s) | Year | Dataset title | Dataset URL | Database, license, and accessibility information |
| --- | --- | --- | --- | --- |
| Joaquina Delás M, Hannon GJ | 2017 | lncRNA dependencies in acute myeloid leukemia | https://www.ncbi.nlm.nih.gov/geo/query/acc.cgi?acc=GSE90072 | Publicly available at the NCBI Gene Expression Omnibus (accession no:GSE90072). Expression data and genome browser tracks available at http://lncrna.hannonlab.org/ |

The following previously published datasets were used:

| Author(s) | Year | Dataset title | Dataset URL | Database, license, and accessibility information |
|---|---|---|---|---|
| Lara-Astiaso D, Weiner A, Lorenzo-Vivas E, Zaretsky I, et al | 2014 | Chromatin state dynamics during blood formation | https://www.ncbi.nlm.nih.gov/geo/query/acc.cgi?acc=GSE59992 | Publicly available at the NCBI Gene Expression Omnibus (accession no: GSE59992) |
| Gazit R, Garrison BS, Rao TN, Shay T, Costello J, Ericson J, Kim F, Collins JJ, Regev A, Wagers AJ | 2013 | Transcriptome Analysis Identifies Regulators of Hematopoietic Stem and Progenitor Cells | https://www.ncbi.nlm.nih.gov/geo/query/acc.cgi?acc=GSE15907 | Publicly available at the NCBI Gene Expression Omnibus (accession no. GSE15907) |
| Roe JS, Mercan F, Rivera K, Pappin DJ, Vakoc CR | 2015 | BET Bromodomain Inhibition Suppresses the Function of Hematopoietic Transcription Factors in Acute Myeloid Leukemia | https://www.ncbi.nlm.nih.gov/geo/query/acc.cgi?acc=GSE66123 | Publicly available at the NCBI Gene Expression Omnibus (accession no: GSE66123) |

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
