## [Decision Letter]

Thank you for submitting your article "lncRNA requirements for mouse acute myeloid leukemia and normal differentiation" for consideration by *eLife*. Your article has been favorably evaluated by Fiona Watt (Senior Editor) and three reviewers, one of whom, Juan Valcárcel (Reviewer #1), is a member of our Board of Reviewing Editors. The following individual involved in review of your submission has agreed to reveal their identity: Jan-Henning Klusmann (Reviewer #3).

The reviewers have discussed the reviews with one another and the Reviewing Editor has drafted this decision to help you prepare a revised submission.

Joint review:

In their study, Delas et al. systematically characterize lncRNAs expression during mouse hematopoiesis and in a well-established mouse model for AML (MLL-AF9/NRAS G12D). Their analysis of previously annotated and de novo assembled lncRNAs led them to uncover several lncRNAs with a specific expression pattern during differentiation and in AML. To investigate the function of 120 lncRNAs with such a specific expression pattern they performed a shRNA in vivo screening, using the same MLL-AF9/NRAS G12D mouse model. They identified 20 lncRNAs whose knock down by at least two hairpins reduces leukemia cell proliferation in vivo, possibly related to the maintenance of a leukemic stem cell state (and inhibition of a myeloid differentiated phenotype). 14 out of these were further validated in one cell line in in vitro competition assays. Mechanistically they linked the function of three of their candidate lncRNAs (Pilna, Pvt1 and Lilam) to the expression of MYC.

These data provide a useful resource for scientists working on hematopoiesis, leukemias and lncRNA biology and provide a proof of principle that specific lncRNAs can play distinct roles in leukemia cell maintenance, in addition to more general functions during hematopoiesis. There have been of course previous efforts to map lncRNAs in both AML and hematopoietic stem cells, as well as reports arguing for a crucial regulatory role for pvt1 in MYC-driven cancers, affecting novelty of some of the data. Nevertheless, the study remains relevant, as it can facilitate understanding the function of lncRNAs in normal and malignant hematopoiesis, which is to date poorly understood. Technically the in vivo screening was thoroughly conducted (including the use of pools of 50 shRNAs) and the candidates well chosen.

Major points:

The manuscript will benefit from addressing issues of validation and generality of the results and from additional functional characterization of the lncRNAs. Specifically:

1) Validation issues.a) Previous studies have demonstrated that lncRNA prediction based on de novo assembly of RNAseq data has a high frequency of false predictions. A recent study showed that the accuracy greatly improved by combining the RNAseq data with CAGE data (Hon et al. Nature 2017). Thus, alignment of the newly identified lncRNAs with the CAGE signals would greatly increase the confidence of their resource.

b) A major caveat for shRNA targeting of lncRNAs is the inefficiency for targeting nuclear lncRNAs due to the fact that the RNAi machinery is mostly cytoplasmic. It would be important to further document the general validity of their shRNA approach to efficiently knockdown nuclear lncRNAs (e.g. are most of the lncRNA candidates biased towards cytoplasmic lncRNA function compared to nuclear roles?). Given the high isoform complexity, the authors should comment to what extent the levels of different regions/isoforms of key lncRNAs are depleted when targeted by shRNAs directed against regions of highest RNAseq coverage.

c) As suggested previously, RNAi has some disadvantages in knocking down lncRNAs, and relying on a single methodology for all the experiments increases the likelihood of false positives. Therefore, the authors should provide an alternative approach to validate the loss of function of these lncRNA candidates, for example antisense oligonucleotides, CRISPRi and paired-guide RNA CRISPR-Cas9 (see Liu et al. Science 2017 and Zhu et al. Nat Biotech 2016).

2) Generality issues.a) They only utilized a single murine cell line model, and it will be beneficial to include another murine AML cell line to exclude cell line-specific bias in these studies. Also, in this paper, the authors claim that Lilam and Pvt1 are AML-specific but have only demonstrated the specificity with one breast cancer cell line; however other groups have shown crucial roles of Pvt1 in ovarian, cervical and colorectal cancer. To claim specificity, the authors should test if knockdown of Lilam and PVT1 is required in other hematopoietic lineages or cell types.

b) PVT1, as previously mentioned is also annotated in human cancers and it would be great to see if this phenomenon is translated into human AMLs (e.g. targeting PVT1 or/and the other lncRNAs in a panel of human MLL-AF9 cell lines).

---

## [Author Response]

Major points:The manuscript will benefit from addressing issues of validation and generality of the results and from additional functional characterization of the lncRNAs. Specifically:1) Validation issues.a) Previous studies have demonstrated that lncRNA prediction based on de novo assembly of RNAseq data has a high frequency of false predictions. A recent study showed that the accuracy greatly improved by combining the RNAseq data with CAGE data (Hon et al. Nature 2017). Thus, alignment of the newly identified lncRNAs with the CAGE signals would greatly increase the confidence of their resource.

CAGE data is unfortunately not available for our mouse cell types. Producing these data and incorporating it into the catalog assembly pipeline would likely take at least a year and would be possible only for a subset of cell types. Others, particularly stem/progenitor populations, are too rare to enable us to obtain sufficient material for conventional CAGE protocols.

Regarding the confidence in our current lncRNA catalog, our assembly is based on two types of RNA sequencing libraries, one of them produced using template switching RT (SMARTer kit from Clontech, see Chenchik et al., 1998). This strategy is specifically designed to capture full-length cDNA. Additionally, we required each individual transcript model to be independently assembled in different samples, dramatically decreasing the likelihood of false predictions.

b) A major caveat for shRNA targeting of lncRNAs is the inefficiency for targeting nuclear lncRNAs due to the fact that the RNAi machinery is mostly cytoplasmic. It would be important to further document the general validity of their shRNA approach to efficiently knockdown nuclear lncRNAs (e.g. are most of the lncRNA candidates biased towards cytoplasmic lncRNA function compared to nuclear roles?). Given the high isoform complexity, the authors should comment to what extent the levels of different regions/isoforms of key lncRNAs are depleted when targeted by shRNAs directed against regions of highest RNAseq coverage.

We appreciate the reviewers’ concerns regarding the potential bias towards cytoplasmic lncRNAs. The main focus of our work was to identify functional lncRNAs where depletion of the RNA had a robust, penetrant phenotype. We therefore felt RNAi was the best approach available, even if we could potentially be biased to cytoplasmic RNAs. Nonetheless, we have tried to address the sub cellular localization of our top candidate lncRNAs by single molecule RNA FISH and sub-cellular fractionations. However, being suspension cells with large nuclei and very little cytoplasma, the RNA FISH was inconclusive, as we could not distinguish perinuclear from true cytoplasmic or nuclear signals. Our multiple attempts at sub cellular fractionation of AML cells, with several protocols, have been unsuccessful, perhaps also due to their unusual nuclear/cytoplasmic ratio.

Regarding the second point, directing the shRNAs towards areas of highest coverage was meant to capture as many isoforms as possible. Consequently, we tend to have more shRNAs in areas that would affect most predicted isoforms. When looking both at qRT-PCR or transcriptome coverage, we can see different transcript areas equally affected. For example, for Pvt1, both hairpins are located in the same area, with shRNA 1 having a stronger knockdown than shRNA 9. However, the level of knockdown is relatively consistent between primer pairs, even though primer set 2 is located spanning two exons ~ 10kb upstream (Author response image 1).

**Author response image 1. respfig1:** (**A**) Pvt1 genomic organization as predicted by our lncRNA catalog and reported on GENCODE, with the shRNA and primer locations indicated. A representative transcriptome coverage plot used for de novo transcriptome assembly is shown. (**B**) qRT-PCR results showing the relative expression of Pvt1 compared to control knockdown (*Renilla*). Two primer pairs are shown on the X-axis. Knockdown with shRNA 1 are shown in blue and with shRNA 9, in purple.

LncRNA_096443 (aka 7358) has a higher expression so we could also look at the overall transcriptome coverage upon knockdown with either shRNA 5 or shRNA 8 (Author response image 2). Although both hairpins are located in the area labeled as C, we can see a coverage reduction also in other areas of the transcript, regardless of the shRNA used.

**Author response image 2. respfig2:** Transcriptome coverage for lncRNA_096443 upon control knockdown (*Renilla*) or knockdown of the lncRNA itself with two independent hairpins, shRNA 5 (light green) and shRNA 8 (dark green). The predicted isoform structures for this lncRNAs are also shown.

c) As suggested previously, RNAi has some disadvantages in knocking down lncRNAs, and relying on a single methodology for all the experiments increases the likelihood of false positives. Therefore, the authors should provide an alternative approach to validate the loss of function of these lncRNA candidates, for example antisense oligonucleotides, CRISPRi and paired-guide RNA CRISPR-Cas9 (see Liu et al. Science 2017 and Zhu et al. Nat Biotech 2016).

We value the reviewers’ concerns regarding false positives. For that reason, we evaluated every phenotype with two independent shRNAs and we carefully checked knock down with different primer sets.

We have additionally made a considerable effort in trying to establish CRISPRi to further validate our results, as suggested by the reviewers. To this end, we transduced our AML cells with a Cas9-KRAB expression vector, which also expresses Cyan Fluorescent Protein (CFP) in a bicistronic fashion (using P2A). We then transduced these cells with a vector expressing sgRNAs against the start site of each lncRNA, against a negative control gene (human olfatory receptor 10A4, OR10A4), or our positive control Replication Protein A3 (RPA3). For RPA3, we used a published sgRNA from Liu et al. Science 2017.

Even after 14 days post sgRNA transduction, the% of sgRNA and dCas9-KRAB expressing cells remained above 50% for the positive control, RPA3 (Author response image 3). This is in stark contrast to our shRNA-mediated proliferation assay where less than 10% of shRNA-expressing cell remain after just two days (Figure 4 of the original paper). Upon analysis of our sgRNA-expressing cells, reductions in transcript levels were very modest (if occurring at all).

Most of our targeted lncRNAs show a slight depletion trend. Given the very mild effect on our positive control, this does not seem surprising. It is possible, though uncertain, that we might eventually optimize CRISPRi in our cell types, but as of now, we have been unsuccessful at applying this approach as a complement to shRNA.

**Author response image 3. respfig3:** Proliferation assay for AML cells expressing dCas9-KRAB and sgRNAs targeting the indicated genes/lncRNAs. Percentage of cells expressing the sgRNA (zsGreen) was monitored over time. Although we made clonal cell line of dCas9-KRAB to ensure they all carry the transgene, we observed silencing of the CFP reporter. For that reason, the gates were set to double positive zsGreen and CFP cells.

2) Generality issues.a) They only utilized a single murine cell line model, and it will be beneficial to include another murine AML cell line to exclude cell line-specific bias in these studies. Also, in this paper, the authors claim that Lilam and Pvt1 are AML-specific but have only demonstrated the specificity with one breast cancer cell line; however other groups have shown crucial roles of Pvt1 in ovarian, cervical and colorectal cancer. To claim specificity, the authors should test if knockdown of Lilam and PVT1 is required in other hematopoietic lineages or cell types.

We have performed proliferation assays following the same set up as Figure 4 (main text) and Figure 4—figure supplement 1 and Figure 5—figure supplement 1 (constitute lncRNA knockdown) in two other AML models (kind gifts from Johannes Zuber), which had not been incorporated to the main text.

Regarding the AML-specificity of Lilam and Pvt1, we did not mean to imply that these lncRNAs had a function specific to AML versus other cancers, rather that their expression was higher in AML than in the rest of the hematopoietic lineages (Figure 7 Main text) and they were required for AML but not for normal myelogenesis (Figure 7 Main text). We have tried to clarify any possible misunderstanding through the text.

b) PVT1, as previously mentioned is also annotated in human cancers and it would be great to see if this phenomenon is translated into human AMLs (e.g. targeting PVT1 or/and the other lncRNAs in a panel of human MLL-AF9 cell lines).

The potentially conserved function of Pvt1 and/or other leukemia lncRNAs in human leukemias is highly interesting. However, establishing these models in the lab would require quite some time. Additionally, lncRNA conservation is not straightforward and these experiments would require identifying the human lncRNAs, designing and validating knockdown tools.